# Evolutionary Origin of *MUTYH* Germline Pathogenic Variations in Modern Humans

**DOI:** 10.3390/biom13030429

**Published:** 2023-02-24

**Authors:** Fengxia Xiao, Jiaheng Li, Philip Naderev Panuringan Lagniton, Si Hoi Kou, Huijun Lei, Benjamin Tam, San Ming Wang

**Affiliations:** 1Ministry of Education Frontiers Science Center for Precision Oncology, Cancer Centre and Institute of Translational Medicine, Department of Public Health and Medical Administration, Faculty of Health Sciences, University of Macau, Macao, China; 2Department of Radiation Oncology, Qilu Hospital of Shandong University, Jinan 250062, China

**Keywords:** *MUTYH*, pathogenic variant, evolutionary origin, phylogenic, archaeological, ancient humans

## Abstract

MUTYH plays an essential role in preventing oxidation-caused DNA damage. Pathogenic germline variations in *MUTYH* damage its function, causing intestinal polyposis and colorectal cancer. Determination of the evolutionary origin of the variation is essential to understanding the etiological relationship between *MUTYH* variation and cancer development. In this study, we analyzed the origins of pathogenic germline variants in human *MUTYH*. Using a phylogenic approach, we searched *MUTYH* pathogenic variants in modern humans in the *MUTYH* of 99 vertebrates across eight clades. We did not find pathogenic variants shared between modern humans and the non-human vertebrates following the evolutionary tree, ruling out the possibility of cross-species conservation as the origin of human pathogenic variants in *MUTYH*. We then searched the variants in the *MUTYH* of 5031 ancient humans and extinct Neanderthals and Denisovans. We identified 24 pathogenic variants in 42 ancient humans dated between 30,570 and 480 years before present (BP), and three pathogenic variants in Neanderthals dated between 65,000 and 38,310 years BP. Data from our study revealed that human *MUTYH* pathogenic variants mostly arose in recent human history and partially originated from Neanderthals.

## 1. Introduction

MUTYH plays a crucial role in preventing oxidation-caused DNA damage. Through glycosylating the mismatched adenine in the OG:A pair, MUTYH guides other components in the base excision repair (BER) pathway to repair the damage to maintain genome stability [1]. Pathogenic germline variations in *MUTYH* damage its function. The biallelic variation in *MUTYH* causes the development of *MUTYH*-associated polyposis (MAP) and colorectal cancer [2]; the monoallelic variation in *MUTYH* primes the carriers to develop colorectal cancer, although the risk is lower than for the biallelic germline variation [3].

*MUTYH* is evolutionarily conserved from bacteria to mammals, reflecting its essential role in maintaining genome stability [4]. Aside from the highly conserved core adenine excision and OG recognition domains [5], *MUTYH* in higher eukaryotes evolved multiple new functional domains, such as the *MUTYH* interdomain connector (IDC) domain to interact with other genes in multiple DNA damage repair (DDR) pathways [6,7]. Human *MUTYH* is under positive selection [8], suggesting that *MUTYH* variants can be beneficial [9], and that there must be rich variants in *MUTYH*. Indeed, 1838 germline variants in human *MUTYH* have been identified and deposited in the ClinVar database [10]. *MUTYH* variants can be deleterious—damaging its function—and 172 germline *MUTYH* variants have been determined as pathogenic, causing MAP and hereditary cancer-predisposing syndrome. For example, biallelic *MUTYH* variants c.527A > G (p.Tyr176Cys) and c.1178G > A (p.Gly393Asp) have been determined as *MUTYH* founder variants for MAP in European populations [11]. 

To better understand human diseases, particularly the pathogenic germline variation-related ones, it requires clarifying the evolutionary origin and the arising time of germline variations. The information will reveal how natural selection selects the deleterious variants, the relationship between adaptations and human diseases, and the heritage of deleterious variations within human population. However, the evolutionary origins of the deleterious variations in human *MUTYH* remain elusive.

In the current study, we explored the evolutionary origins of pathogenic variants in human *MUTYH*. While the conservation of human *MUTYH* sequences across different species suggests that human *MUTYH* pathogenic variants could have originated from common ancestors, the positive selection in human *MUTYH* suggests that human *MUTYH* pathogenic variants may have also arisen during the human evolution process. The rapid development of genomics has provided abundant genome sequences from various species, ancient humans, and the extinct Neanderthals and Denisovans [12]. Taking advantage of the rich genome resources, we conducted both phylogenic and archaeological studies to investigate the evolutionary origin of the human *MUTYH* pathogenic variants. Our study reveals that human *MUTYH* pathogenic variants mainly arose in recent human history and were partially inherited from Neanderthals.

## 2. Materials and Methods

### 2.1. Source of Human MUTYH Variants 

Human *MUTYH* variants were downloaded from the ClinVar database [10]. Based on the classes, the variants were divided into the pathogenic variants (PVs) group including “Pathogenic,” “Likely Pathogenic” and “Pathogenic and Likely Pathogenic”; and the benign variants (BVs) group including “Benign,” “Likely Benign” and “Benign and Likely Benign”. Single-nucleotide variants and indels affecting <4 bases were included in the study.

### 2.2. Phylogenetic Analysis

The process followed the detailed procedures described in our previous publication [13]. Briefly, the reference sequences used in annotating human *MUTYH* variants were: hg38 NC_000001.11 for the genomic position, NM_012222 for cDNA and NP_036354 for protein. *MUTYH* genomic sequences included 100 species of vertebrates in eight clades: Primate, Euarchontoglires, Laurasiatheria, Afrotheria, Mammal, Aves, Sarcopterygii and Fish. Sequence alignment was performed following the procedures in Multiz Alignments in the UCSC genome browser [14]. The PhyloFit program in the PHAST package was used to build the tree model for the 100 vertebrate species [15]. PhastCons and phyloP in the PHAST package were used to measure evolutionary conservation. Lastz (BLASTZ) and Multiz were used to align repeat-masked sequences between human hg38 and non-human *MUTYH* genome sequences [16,17,18,19]. The scoring matrix and pairwise parameters for each species were adjusted by referring to the phylogenetic distances from the references. GetBase [20] was used to obtain the bases at the positions corresponding to human pathogenic and benign variants.

### 2.3. Archaeological Analysis

Ancient human genome information was collected from Allen Ancient DNA Resource [21], PubMed and Google scholar. Afterwards, BAM or SRA files of ancient human DNA sequences and related publications were downloaded from the European Nucleotide Archive [22], the Max Planck Institute Genome Projects [23] and the National Genomics Data Center [24]. The sequences containing *MUTYH* (chr1:45794835–45806142, hg19 and Chr1: 45567501–45578729, hg18 by Ensembl) were extracted using the sam-dump command in the official SRA toolkit, and base quality scores were rescaled using mapDamage 2.0 [25]. The *MUTYH* sequences were mapped to hg19 or hg18 reference sequences depending on the versions of the ancient DNA sequences. The Mpileup command in SAMtools [26] was used for variant calling from the mapped sequences with the minimal base quality of 1 [26]. After generating VCF files of ancient humans, the called variants were annotated with ANNOVAR [27] using the table_annovar.pl script containing the ClinVar, refGene and avsnp150. The variants from ancient DNA were compared with those from ClinVar to identify the shared and unshared variants between ancient and modern humans. For the shared variants, the locations and the fossil ages of the ancient carriers were identified from their original publications. The geographical distribution map of the ancient *MUTYH* pathogenic variants was created using Matlab software (The MathWorks, Inc., Natick, MA, USA). The following conditions were set to ensure the reliability of the variants identified in the ancient sequences: (1) MapDamage was used to remove false-positive variants caused by deamination of ancient DNA; (2) the called variants were manually checked for their reliability; (3) each variant must have a dbSNP ID to diminish sequencing artefacts.

### 2.4. Statistical Analysis

Comparison of sharing rates between PV and BV groups was performed by Mann–Whitney U test using SPSS software version 24 (*p* < 0.05 as significant).

## 3. Results

### 3.1. Phylogenetic Analysis of Human MUTYH Variants in Non-Human Vertebrates

A total of 750 germline variants in human *MUTYH* consisting of 172 PVs and 578 BVs were included in the study (Appendix A). The variants were searched in the 99 vertebrates in 8 clades of Primate, Euarchontoglires, Laurasiatheria, Afrotheria, Mammal, Aves, Sarcopterygii and Fish. The results were as follows:

#### 3.1.1. PVs

Thirty-three (19.2%) of the 172 human *MUTYH* PVs were identified in 36 species, of which stop-gain/nonsense variants, nonsynonymous SNV, splicing and frameshift deletion accounted for 48.5%, 21.2%, 15.2% and 12.1%, respectively (Figure 1, Appendix A). We observed the following features of the shared variants.

Species in Primate barely shared human PVs. Species in Primate are the closest relatives to humans. However, there were no human *MUTYH* PVs shared in the chimp, gorilla, gibbon, rhesus, crabeating macaque, baboon, green monkey, marmoset, squirrel monkey or bushbaby. There were two human PVs of c.760del (p.Val255Phefs) and c.499del (p.Glu166_Val167insTer) shared with orangutans. Mice and rats are the widely used animal models in biomedical studies, but they did not share any human *MUTYH* PVs. In contrast to Primate, the distant clade Aves had the highest number of shared PVs of 14, and Fish had the second highestof 12 (Appendix A, Appendix A). c.931C > T (p.Gln311Ter) was the most shared, being found in six species, including the lesser Egyptian jerboa, Chinese hamster and golden hamster among Euarchontoglires; the black flying fox and megabat in Laurasiatheria; and zebrafish in Fish. The three haplotype-confirmed human *MUTYH* founder mutations of c.527A > G (p.Tyr176Cys) and c.1178G > A (p.Gly393Asp) in the European population [11] and c.924 + 3A > C in the Italian population [28] were absent in all non-human vertebrates.

Shared variants did not follow the evolutionary tree. Of the 33 shared PVs, 23 (69.7%) were shared with only 1 species. For the variants shared by multiple species, they did not follow the order of the evolutionary tree continuously from the closest to the most distant from humans (Appendix A, Appendix A). For example, the six species in three clades indicated above shared c.931C > T (p.Gln311Ter); the pig in Laurasiatheria and rock pigeon in Aves shared c.682 − 1G > A; the platypus in Mammal and painted turtle in Sarcopterygii shared c.323T > C (p.Leu108Pro); and the Tasmanian devil and wallaby in Mammal and stickleback in Fish shared c.1231C > T (p.Gln411Ter).

#### 3.1.2. BVs

A total of 519 (90% of 578) of the human BVs were shared with 98 species across all 8 clades (Appendix A, Figure 1B, Appendix A), indicating *MUTYH* BVs had a distinct sharing pattern from PVs.

Primates shared a total of 295 human BVs. For example, chimps shared eight human BVs (5 intronic: c.36 + 325C > G, c.340 − 7T > C, c.495 + 35A > G, c.1468 − 40C > G, c.1510 − 14C > G; 3 exonic: c.999T > C (p.Thr333=), c.1347A > G (p.Thr449=), c.1422G > C (p.Thr474=)). Mice shared the most (100) human BVs. At the clade level, Laurasiatheria had the highest sharing numberof 1677 BVs. The Mann–Whitney U test showed that the sharing rates of human BVs and human PVs were significantly different (*p* < 0.05) (Appendix A).

The shared BVs followed the evolutionary tree. Ten BVs were shared by all 8 clades, 24 were shared by 7 clades and 47 were shared by 6 clades, although Primate had a lower sharing rate than other clades (Appendix A, Figure 1B). All species but lamprey in Fish shared human BVs. Most of the sharing continuously followed the evolutionary order (Appendix A).

The results from the phylogenetic analysis demonstrated that evolutionary conservation was part of the origin of human *MUTYH* BVs, but not the origin for human *MUTYH* PVs.

### 3.2. Archaeological Analysis of MUTYH Variants in Ancient Humans

Using the same set of human *MUTYH* variants used in the phylogenetic analysis, we tested whether human *MUTYH* variants could arise during human evolution. We searched the variants in 5031 ancient humans dated between 45,045 and 100 years BP, 26 Neanderthals dated between 120,000 and 38,310 years BP, and 4 Denisovans dated between 158,500 and 69,650 years BP. 

#### 3.2.1. PVs

A total of 24 human *MUTYH* PVs were identified in 42 ancient human individuals dated between 30,570 and 480 years BP. We observed the following features:

Most of the shared PVs arose recently. Except c.55C > T (p.Arg19Ter) in a carrier in Dryanovo, Bulgaria, dated to 30,570 years BP [29], 41 out of the 42 relevant ancient human individuals were dated to have lived within the last 10,000 years (Table 1, Figure 2); the youngest carrier for c.316C > T (p.Arg106Trp) in Atajadizo, Dominican Republic, was dated to 480 years BP [30].

Stop-gain variants were common in the shared PVs. Of the 24 shared *MUTYH* PVs, 14 (58.3%) were stop-gain variants. 

Shared PVs were located in different domains. The top three clustered domains were EndoIII-iron–sulfur cluster (FeS)-like domain, EndoIII-6-Helix_barrel domain and Nudix-like domain (Figure 3). 

The shared PVs were highly reliable. Of the 24 shared *MUTYH* PVs, 11 were present in multiple individuals including 6 PVs in 2 individuals, 3 PVs in 3 individuals, 1 PV in 4 individuals and 1 PV in 5 individuals (Table 2). The presence of dbSNP ID for each variant further enhanced the reliability of the shared PVs.

Founder variants were shared. The 24 shared PVs included two *MUTYH* founder variants of c.527A > G (p.Tyr176Cys) and c.1178G > A (p.Gly393Asp) in the European population [11]. c.536A > G (p.Tyr179Cys) was present in two individuals: one individual dated to 7701 years BP from Italy and the other dated to 5730 years BP from England; c.1187G > A (p.Gly396Asp) was present in three individuals: one dated to 8315 years BP from Turkey, one dated to 3713 years BP from Kazakhstan and another dated to 2185 years BP from Czech Republic.

Neanderthals shared *MUTYH* PVs with modern humans. Three *MUTYH* PVs were identified in three Neanderthals dated from 65,000 to 38,310 years BP. c.1205C > T (p.Pro402Leu) identified in Neanderthals was also detected in ancient humans. No human *MUTYH* PV was detected in Denisovans (Table 1 and Table 2).

#### 3.2.2. BVs

A total of 126 human *MUTYH* BVs (Appendix A) were identified in 2217 ancient humans dated from 37,470 to 135 years BP. These shared BVs had the following features:

*MUTYH* BVs were highly shared between ancient and modern humans. Twenty-two (17.5%) were present in 2 ancient individuals, 10 (7.9%) were present in 3 individuals and 18 (14.3%) were present in multiple individuals. c.495 + 35A > G had the highest sharing within 1821 carriers, and c.1468 − 40C > G had the second highest sharing within 516 carriers (Appendix A).

Most BVs arose recently. Except for c.1468 − 40C > G in a carrier in Russia dated to 37,470 years BP [51], most of the shared BVs arose within the last 10,000 years. c.495 + 35A > G was the youngest one identified in a carrier in Efate, Vanuatu, dated to 135 years BP [52].


Shared BVs were mostly synonymous (72, 57.1%) and intronic (44, 34.9%).


Neanderthals and Denisovans shared *MUTYH* BVs with modern humans. A total of 31 BVs were identified in the Neanderthals, and 7 were identified in Denisovans. Of these 38 BVs, 32 were synonymous and intronic (Table 2, Appendix A). c.1468 − 40C > G was shared by ancient humans, Neanderthals and Denisovans; 18 BVs were shared by two groups, of which 1 was shared between Denisovans and Ancient humans; and 17 were shared between Neanderthals and ancient humans (Appendix A).

The results from the phylogenical and archaeological analysis showed that human *MUTYH* PVs mostly originated from ancient humans and partly from the extinct Neanderthals. In contrast, human *MUTYH* BVs mostly originated from non-human vertebrates or humans and partly from the extinct Neanderthals and Denisovans.

## 4. Discussion

By referring to the rich genomic data from non-human vertebrate species and ancient humans, our study reveals that human *MUTYH* PVs mostly originated during recent human history.

Data from our phylogenetic study in non-human vertebrates do not support cross-species evolutionary conservation as the source for human *MUTYH* PVs. This is reflected by the fact that almost no human PVs were shared in the Primate clade. For the non-human species sharing human *MUTYH* PVs, they were mostly the species in Aves and Fish clades distant from humans. While these shared PVs could likely be generated by chance in humans and these shared species, the cause for the sharing of genetic variants with distant species remains unclear. Different theories have been proposed in trying to explain the situation, such as the “founder effect,” “fixations of slightly deleterious mutations,” “relaxed selection on late-onset phenotypes” and “compensatory changes” [53]. The “compensatory changes” theory is considered a favorable explanation for the sharing of human PVs with distant species. It considers that “compensatory mutations at other sites of the same or a different protein render the deleterious mutations neutral”. In comparison, human *MUTYH* BVs shared with other vertebrates followed the order of the evolutionary tree, which served as a convincing control for PVs’ sharing pattern, which did not follow the order of the evolutionary tree.

Our anthropological analysis of ancient humans identified 42 ancient carriers for 24 human *MUTYH* PVs, including two founder variants. Forty-one out of the 42 ancient carriers were dated between 9100 and 480 years BP, highlighting that human *MUTYH* PVs likely originated during recent human history after the last glacier period of the Quaternary that ended approximately 11,000 years BP. Pathogenic variants are deleterious for fitness. Therefore, evolutionary selection should suppress their presence in the human population. However, multiple *MUTYH* PVs were present at rather high levels, suggesting their potential benefitial effects. While we do not have enough evidence to support this possibility, we speculate that the pathogenic effects could be developmental stage-dependent: the pathogenic variants could be beneficial at the reproductive stage but deleterious at later reproduction stages. Cancer caused by the pathogenic variants occurs mostly at the later reproduction stage. Before reaching the stage, however, the PVs have already been transmitted to the next generation. The abundance of BVs shared across many species indicates the beneficial or neutral effects. Only 24 out of the 172 *MUTYH* PVs were identified in ancient humans The incidence of 0.85% implies that 1 out of 117 ancient individuals carried one *MUTYH* germline pathogenic variant, which is lower than the carrier frequency in modern humans estimated to be around 1:45 [54]. We speculate that the origination of *MUTYH* PVs has been gradually increasing following the expansion of the human population. 

It is interesting to note that most of the *MUTYH* PVs were present in modern humans but not in ancient humans since only 24 out of the 172 *MUTYH* PVs were identified in ancient humans. The following factors could contribute to the fact: (1) Different sizes of ancient and modern human populations. In this study, only 24 out of the 172 *MUTYH* PVs from modern humans were identified in ancient humans. It was estimated that there were only a few hundred to a few thousand individuals when human migrated out of Africa. Afterwards, the size of the human population gradually increased but rapidly expanded after the last glacier period (11,000 years BP), and further increased after agricultural development (7000–5000 years BP) till nowadays. The probability of new variants arising in a larger population, therefore, increased. (2) The limited number of ancient samples. Although a significant progress in anthropological genomics has been made, genomic sequence data from ancient humans remain limited that they are only available from about 5000 ancient individuals. With data from more ancient samples, it would be expected that more PVs found in modern humans could be identified in ancient humans. (3) Different lifespans for modern and ancient humans. Modern humans have much greater longevity than ancient humans did. Cancer is an aging disease. With prolonged lifespans, cancer incidence in the modern human population is increasing as shown by epidemiological data. Although evidence indeed revealed the presence of cancer in ancient humans, only a few PVs were detected in ancient individuals.

Anthropological genomic studies found that parts of the modern human genome were inherited from Neanderthals and Denisovans [55,56,57], and some alleles were related to human diseases, such as a risk for depression and skin lesions [58]. Consistent with that observation, our study identified multiple *MUTYH* PVs and BVs in Neanderthals and Denisovans, highlighting that the extinct Neanderthals and Denisovans also contributed disease susceptibility, including cancer, to modern humans. 

Our findings are consistent with the observation that most deleterious protein-coding single nucleotide variants (SNVs) arose in the past few thousand years [59]. Through direct comparison of variants between modern humans and vertebrates and ancient humans, our study provide further evidence to support the concept that pathogenic variants in human disease-related genes could largely arise in recent human history.

## Figures and Tables

**Figure 1 biomolecules-13-00429-f001:**
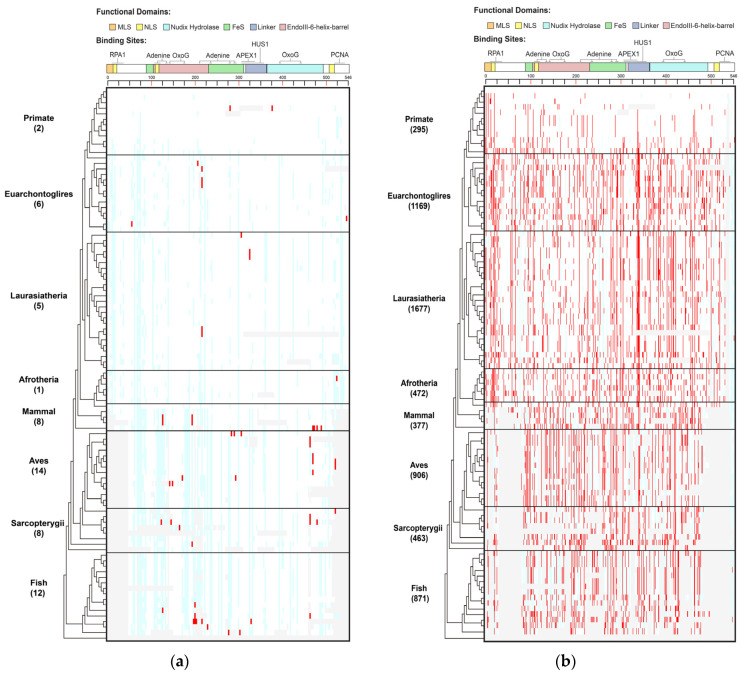
Distribution of human *MUTYH* variants in 100 vertebrates of 8 clades. (**a**) Human *MUTYH* PVs; (**b**) Human *MUTYH* BVs. White cell: same as human wild type; red cell: same as human variants; light blue cell: different from both human wild type and variants; gray cell: gaps or unaligned. *X* axis: the shared variants, *Y* axis: the 100 species from human in Primate to lamprey in Fish. The schematic diagram of *MUTYH* functional domains and binding sites is shown on top of the figures.

**Figure 2 biomolecules-13-00429-f002:**
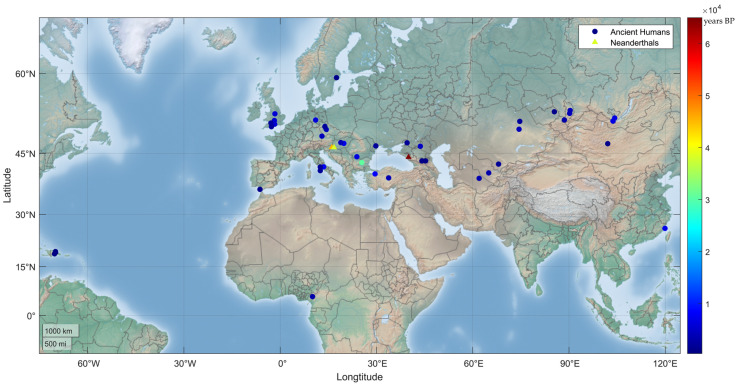
Geographic distribution of human *MUTYH* PVs in ancient humans. A total of 24 human *MUTYH* PVs were identified in 42 ancient individuals, mostly from Europe and Asia, dated from 30,570 to 480 years BP. One of the 42 individuals had 2 PVs, and each of the other 41 individuals had 1 PV. Three PVs were also identified in the Neanderthals (triangle dots) of Croatia and Russia, dated between 65,000 and 38,310 years BP.

**Figure 3 biomolecules-13-00429-f003:**
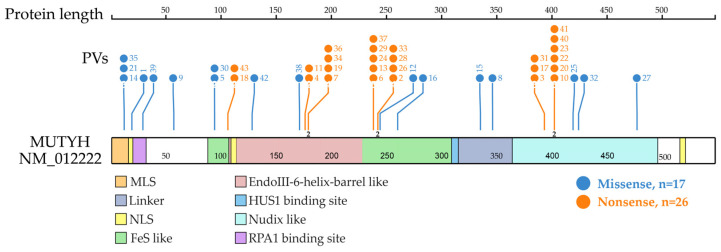
Distribution of *MUTYH* PVs shared in ancient individuals. It shows 21 codon positions of the 24 *MUTYH* PVs detected in 42 ancient individuals, of which 23 were located at functional domains. Each line represents one codon position with the PV number at the top (same order as in Table 1 dating from the oldest to the newest). Each codon position had one variant, except for 3 positions that had 2 variants represented by the number “2” at the bottom of the line. The top 3 clustered functional domains were EndoIII-iron–sulfur cluster (FeS)-like (15 PVs), EndoIII-6-Helix_barrel (12 PVs) and Nudix-like (12 PVs). Other domains included replication protein A1 (RPA1) binding site (5 PVs), mitochondrial localization signal (MLS) (3 PVs), HUS1 binding site (2 PVs), linker (2 PVs) and nuclear localization signal (NLS) (1 PV).

**Table 1 biomolecules-13-00429-t001:** List of *MUTYH* PVs shared in ancient individuals.

Order	Years (BP)	Fossil Site	Variation		Type	dbSNP	Functional Domain [31]	Reference
			cDNA	Protein				
**A.Ancient Human**								
1	30,570	Dryanovo, Bulgaria	c.55C > T	p.Arg19Ter	Stopgain	rs587780088	NLS, RPA1 Binding site	[29]
2	9100	United Kingdom	c.724C > T	p.Arg242Cys	Nonsynonymous SNV	rs200495564	FeS like	[32]
3	8299	Bartin, Turkey	c.1178G > A *	p.Gly393Asp	Nonsynonymous SNV	rs36053993	Nudix like	[33]
4	7628	Grotta Continenza, Italy	c.527A > G *	p.Tyr176Cys	Nonsynonymous SNV	rs34612342	EndoIII-6-Helix-barrel like	[34]
5	7500	Matsu archipelago, China	c.280C > T	p.Arg94Ter	Stopgain	rs138775799	FeS like, EndoIII-6-Helix-barrel like	[35]
6	7033	Hungary	c.712C > T	p.Arg238Trp	Nonsynonymous SNV	rs34126013	FeS like	[33]
7	6713	Irkutsk, Russia	c.535C > T	p.Arg179Cys	Nonsynonymous SNV	rs747993448	EndoIII-6-Helix-barrel like	[36]
8	6700	Irkutsk, Russia	c.1038G > A	p.Trp346Ter	Stopgain	rs1060501324	Linker, HUS1 binding site	[36]
9	6470	Oltenia, Romania	c.169G > T	p.Glu57Ter	Stopgain	rs1557487793	Not a domain	[37]
10	6211	Germany	c.1205C > T	p.Pro402Leu	Nonsynonymous SNV	rs529008617	Nudix like	[38]
11	5656	United Kingdom	c.527A > G *	p.Tyr176Cys	Nonsynonymous SNV	rs34612342	EndoIII-6-Helix-barrel like	[32]
12	5610	United Kingdom	c.730C > T	p.Arg244Ter	Stopgain	rs587782885	FeS like	[32]
13	5455	Gloucestershire, England	c.712C > T	p.Arg238Trp	Nonsynonymous SNV	rs34126013	FeS like	[39]
14	4649	Russia	c.35G > A	p.Trp12Ter	Stopgain	rs1064795596	MLS, RPA1 Binding site	[40]
15	4249	Germany	c.1003C > T	p.Gln335Ter	Stopgain	rs587780082	Linker, HUS1 binding site	[40]
16	4239	Uybat, Russia	c.780G > A	p.Trp260Ter	Stopgain	rs1338038953	MSH6 binding site	[36]
17	4100	Kazakhstan	c.1178G > A *	p.Gly393Asp	Nonsynonymous SNV	rs36053993	Nudix like	[36]
18	4078	Verkhni Askiz, Russia	c.316C > T	p.Arg106Trp	Nonsynonymous SNV	rs765123255	FeS like, EndoIII-6-Helix-barrel like	[36]
19	3850	Czech Republic	c.535C > T	p.Arg179Cys	Nonsynonymous SNV	rs747993448	EndoIII-6-Helix-barrel like	[41]
20	3774	Hungary	c.1204C > T	p.Pro402Ser	Nonsynonymous SNV	rs121908382	Nudix like	[42]
21	3600	Kaman, Turkey	c.35G > A	p.Trp12Ter	Stopgain	rs1064795596	MLS, RPA1 Binding site	[36]
22	3407	Kazburun, Turkmenistan	c.1205C > T	p.Pro402Leu	Nonsynonymous SNV	rs529008617	Nudix like	[43]
23	3145	Uzbekistan	c.1205C > T	p.Pro402Leu	Nonsynonymous SNV	rs529008617	Nudix like	[44]
24	3099	Russia	c.712C > T	p.Arg238Trp	Nonsynonymous SNV	rs34126013	FeS like	[40]
25	3065	Shum Laka, Cameroon	c.1255C > T	p.Gln419Ter	Stopgain	rs1437789978	Nudix like	[42]
26	2320	Laos	c.724C > T	p.Arg242Cys	Nonsynonymous SNV	rs200495564	FeS like	[45]
27	2248	United Kingdom	c.1429G > T	p.Glu477Ter	Stopgain	rs121908381	Nudix like	[41]
28	2234	Glinoe, Moldova	c.725G > A	p.Arg242His	Nonsynonymous SNV	rs140342925	FeS like	[43]
29	2203	Saryarka, Kazakhstan	c.712C > T	p.Arg238Trp	Nonsynonymous SNV	rs34126013	FeS like	[46]
30	2200	United Kingdom	c.280C > T	p.Arg94Ter	Stopgain	rs138775799	FeS like, EndoIII-6-Helix-barrel like	[41]
31	2185	Czech Republic	c.1178G > A *	p.Gly393Asp	Nonsynonymous SNV	rs36053993	Nudix like	[41]
32	1933	Rostov, Russia	c.1272G > A	p.Trp424Ter	Stopgain	rs1060501325	Nudix like	[46]
33	1850	Via Paisiello, Italy	c.725G > A	p.Arg242His	Nonsynonymous SNV	rs140342925	FeS like	[34]
34	1850	Via Paisiello, Italy	c.535C > T	p.Arg179Cys	Nonsynonymous SNV	rs747993448	EndoIII-6-Helix-barrel like	[34]
35	1804	South Kazakhstan	c.35G > A	p.Trp12Ter	Stopgain	rs1064795596	MLS, RPA1 Binding site	[46]
36	1236	Arkhangai, Mongolia	c.536G > A	p.Arg179His	Nonsynonymous SNV	rs143353451	EndoIII-6-Helix-barrel like	[47]
37	1208	Russia	c.712C > T	p.Arg238Trp	Nonsynonymous SNV	rs34126013	FeS like	[40]
38	1125	Cadiz, Spain	c.513G > A	p.Trp171Ter	Stopgain	rs1570423722	EndoIII-6-Helix-barrel like	[30]
39	1050	Birka, Sweden	c.85C > T	p.Gln29Ter	Stopgain	rs768386527	RPA1 Binding site	[48]
40	850	Atajadizo, Dominican Republic	c.1205C > T	p.Pro402Leu	Nonsynonymous SNV	rs529008617	Nudix like	[30]
41	850	North Ossetia	c.1204C > T	p.Pro402Ser	Nonsynonymous SNV	rs121908382	Nudix like	[46]
42	850	North Ossetia	c.384G > A	p.Trp128Ter	Stopgain	rs587781295	EndoIII-6-Helix-barrel like	[46]
43	480	Atajadizo, Dominican Republic	c.316C > T	p.Arg106Trp	Nonsynonymous SNV	rs765123255	FeS like, EndoIII-6-Helix-barrel like	[30]
B.Neanderthals								
1	65,000	Sukhoi Kurdzhips, Russia	c.848G > A	p.Gly283Glu	Nonsynonymous SNV	rs730881833	FeS like, Adenine Binding site	[49]
2	45,500	Donja Voca, Croatia	c.1205C > T	p.Pro402Leu	Nonsynonymous SNV	rs529008617	Nudix like	[49]
3	38,310	Donja Voca, Croatia	c.679C > T	p.Gln227Ter	Stopgain	rs1064796630	EndoIII-6-helix-barrel like	[50]

* Founder mutation; FeS: EndoIII-iron–sulfur cluster; MLS: mitochondrial localization signal; RPA1: replication protein A1; NLS: nuclear localization signal.

**Table 2 biomolecules-13-00429-t002:** Summary of *MUTYH* variants shared in ancient individuals.

Category		Ancient Samples (%)	
	Ancient humans	Neanderthals	Denisovans
**PVs**			
Types of variants			
Stopgain	14 (58.3)	1 (33.3)	-
Nonsynonymous SNV	10 (41.7)	2 (66.7)	-
Total variants	24 (100)	3 (100)	-
Variants shared by			
1 carrier	13 (54.2)	3 (100)	-
2 carriers	6 (25.0)	-	-
3 carriers	3 (12.5)	-	-
4 carriers	1 (4.2)	-	-
5 carriers	1 (4.2)	-	-
Total variants	24 (100)	3 (100)	-
Total carriers	42	3	-
**BVs**			
Types of variants			
Synonymous SNV	72 (57.1)	12 (38.7)	4 (57.1)
Intronic SNV	44 (34.9)	15 (48.4)	1 (14.3)
Nonsynonymous SNV	4 (3.2)	2 (6.5)	-
UTR	4 (3.2)	2 (6.5)	2 (28.6)
Deletion	1 (0.8)	-	-
Total variants	126 (100)	31 (100)	7 (100)
Variants shared by			
1 carrier	76 (60.3)	19 (61.3)	7 (100)
2 carriers	22 (17.5)	4 (12.9)	-
3 carriers	10 (7.9)	1 (3.2)	-
4 carriers	2 (1.6)	1 (3.2)	-
5 carriers	2 (1.6)	1 (3.2)	-
7 carriers	1 (0.8)	1 (3.2)	-
11 carriers	1 (0.8)	1 (3.2)	-
12 carriers	1 (0.8)	1 (3.2)	-
14 carriers	1 (0.8)	1 (3.2)	-
17 carriers	1 (0.8)	1 (3.2)	-
21 carriers	1 (0.8)	-	-
29 carriers	1 (0.8)	-	-
163 carriers	1 (0.8)	-	-
166 carriers	1 (0.8)	-	-
183 carriers	1 (0.8)	-	-
276 carriers	1 (0.8)	-	-
382 carriers	1 (0.8)	-	-
516 carriers	1 (0.8)	-	-
1821 carriers	1 (0.8)	-	-
Total variants	126 (100)	31 (100)	7 (100)
Total carriers	2217	18	1

## Data Availability

Publicly available datasets were analyzed in this study, which can be found through references [10,14,21,22,23,24].

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
