# Peer review of "Evolutionary Origin of MUTYH Germline Pathogenic Variations in Modern Humans"

_biomolecules, 2023, doi:10.3390/biom13030429_

Round 1
Reviewer 1 Report
Dear authors,
Thank you very much for investigating this subject. The manuscript has originality and sound scientific results. It would have been interesting if there was a speculation why, although cancer was present in early humans, these pathogenic variations are only present in modern humans. Other than that I have nothing to add to this extensive paper. Thanks for adding to the limited knowledge on this subject.
Kind regards,
The reviewer
Author Response
Thank you very much for the comments and raising a very interesting question.
We consider several possible reasons for “why these pathogenic variations are only present in modern humans although cancer was present in early humans”:
1) The different sizes between ancient and modern human population. In this study, only 24 out of the 172 MUTYH PVs from modern humans were identified in ancient humans. It was estimated that there were only a few hundreds to thousands of individuals when the human migrated out of Africa. Afterwards, the size of human graduatedlly increased but rapidly expanded after the last glacier period (11,000 years before present), and further increased after agricultural development (7,000-5,000 years before present) till nowadays. The probability of new variants arising in the larger population size therefore increases;
2) The limited number of ancient samples. Although a significant progress in recent anthropological study is the anthropological genomics, genomic sequence data remain limited with only about 5,000 ancient individuals available comparing to the data from modern humans. With more data from more ancient samples, it would be expected that more pathogenic variants in modern humans could be identified in the ancient humans.
3) Different lifespans between modern and ancient humans. Modern human has much longer longevity than ancient human. Cancer is an aging disease. With the prolonger lifespans, cancer incidence in human population increases as shown by population epidemiological data although evidence indeed revealed the presence of cancer in ancient human and a few pathogenic variations were detected in ancient individuals.
In the revision, the following paragraph has been added under Discussion to address the issue:
It is interesting to note that most of the pathogenic variants were present in modern human but not in ancient human. We consider that the following factors could contribute to the fact: 1) The different sizes between ancient and modern human population. In this study, only 24 out of the 172 MUTYH PVs from modern humans were identified in ancient humans. It was estimated that there were only a few hundreds to thousands of individuals when the human migrated out of Africa. Afterwards, the size of human gradually increased but rapidly expanded after the last glacier period (11,000 years before present), and further increased after agricultural development (7,000-5,000 years before present) till nowadays. The probability of new variants arising in the larger population size therefore increases; 2) The limited number of ancient samples. Although a significant progress in recent anthropological study is the anthropological genomics, genomic sequence data remain limited with only about 5,000 ancient individuals available compared to the data from modern humans. With more data from more ancient samples, it would be expected that more pathogenic variants in modern humans could be identified in the ancient humans; 3) Different lifespans between modern and ancient humans. Modern human has much longer longevity than ancient human. Cancer is an aging disease. With the prolonger lifespans, cancer incidence in modern human population increases as shown by population epidemiological data although evidence indeed revealed the presence of cancer in ancient human and a few pathogenic variations were detected in ancient individuals.
Reviewer 2 Report
The study conducted by Xiao and colleagues aims to analyzed the origin of pathogenic germline variants in human MUTYH gene - a well-known DNA repair gene. Mutations in this gene are responsible for MUTYH-associated polyposis (MAP), an autosomal recessive disorder characterized by predisposition for adenomatous polyps in the colon. Knowledge about origin of these mutations in human population is limited. The Authors performed analysis of distribution MUTYH mutation in human and in 99 vertebrates from eight clades as well as in ~ 5,000 ancient human genomes. From methodological point of view, the study is correctly conducted. However, the limitation of the study is a small size of studied populations (in term of geographical region and time of origin). Obtained results based on simple DNA sequence comparisons from available genomics data collection. No new data were obtained in this study. Moreover, the paper is very similar (the same scheme, tables, figures) to previous paper (doi: 10.26508/lsa.202101263) of the Authors, where BRCA1 and BRCA2 genes were analyzed. Thus, originality and innovation of the present work is very limited (next gene, the same research approach).
I recommend the resubmission of the manuscript to journal which focus more on genomic structure (eg. DNA, CIMB).
Author Response
Thank you for the review comments and the questions are answered one by one below:
Question:
I recommend the resubmission of the manuscript to journal which focus more on genomic structure (eg. DNA, CIMB).
Answer:
The other two reviewers provide favorable comments for our study to cells.
Question:
The limitation of the study is a small size of studied populations (in term of geographical region and time of origin).
Answer:
Although a significant progress in recent anthropological study is the anthropological genomics, sequence data from ancient human remain limited. By extensive searching in the published anthropological genomics data sources, we tried to maximize the ancient genomics data to increase the size of ancient human population for our study with about 5,000 ancient individuals available originated from different geographical regions and time scale. With more data from more ancient samples, it would be expected that more pathogenic variants in modern humans could be identified in the ancient humans. In recognizing the limitation, we stated in Discussion “the small sharing number can also be related to the limited sample size of the 5,031 ancient human samples included in the study. Analyzing more ancient human samples while available may identify more MUTYH PVs shared between modern and ancient humans”. After revision, the description was inserted in a new paragraph as “2) The limited number of ancient samples. Although a significant progress in recent anthropological study is the anthropological genomics, genomic sequence data remain limited with only about 5,000 ancient individuals available compared to the data from modern humans. With more data from more ancient samples, it would be expected that more pathogenic variants in modern humans could be identified in the ancient humans”.
Question:
Obtained results based on simple DNA sequence comparisons from available genomics data collection. No new data were obtained in this study.
Answer:
Our study relies on genome sequences from vertebrates and ancient human. A major goal of genome sequencing study is to provide reference sequences for biomedical studies. There is no need for every study to generate de novo genome sequences. For example, there is no need to sequence all ancient human samples as it is not possible for every lab to collect and sequence the ancient DNA samples, which has become a highly specialized subject. Instead, mining the published data from different anchors is the way to go. For example, there is no need to sequence thousands of human genomes for human variant study, as tens of thousands of human genome data are already publically available for the study. This is the beauty of genomic study.
Question:
Moreover, the paper is very similar (the same scheme, tables, figures) to previous paper (doi: 10.26508/lsa.202101263) of the Authors where BRCA1 and BRCA2 genes were analyzed. Thus, originality and innovation of the present work is very limited (next gene, the same research approach).
Answer:
Our previous study in BRCA1 and BRCA2 was our first taste in systematic investigation for the evolutionary origin of tumor suppressor genes. However, there are hundreds of genes involving DNA damage repair function. While each gene involves in different pathways and plays different roles in DNA damage repair, their evolution origins remain largely unclear. Although our current study used similar approach to our previous BRCA1 study, the biological significance between MUTYH and BRCA is different: BRCA involves in homolog recombination pathway for double strand damage repair, whereas MUTYH involves in oxidation-caused DNA damage repair.
Reviewer 3 Report
This is a well written presentation of an interesting and important study of the evolutionary origin of MUTYH germline mutations in humans. I have only minor suggestions. 1. The incidence of pathogenic variations in modern humans should be stated and compared to the findings in ancient humans. 2. The life expectancy of ancient humans is estimated to have been about 38 years, approximately half as long as modern humans and the average age of presentation of colorectal cancer is 50 years of age. How could this difference between the study groups have affected the data (if at all)? 3. The authors speculate (line 272) that pathogenic variations beneficial in promoting survival and reproduction. This statement should be justified as to how PVs can be beneficial.
approximately
approximately
half as long as life expectancy.
Author Response
Thank you very much for the comments and raising very interesting questions. The followings are my answers to each question:
Question:
The incidence of pathogenic variations in modern humans should be stated and compared to the findings in ancient humans.
Answer:
In the revision the following sentences have been included in Discussion:
… and the incidence was 0.85%, meaning that 1 out of 117 ancient individuals carried one germline pathogenic MUTYH variant, lower than the carrier frequency in modern humans, which was estimated to be around 1:45 [57]. We speculate that the origination of PVs is gradually increasing following the expansion of human population.
Question:
The life expectancy of ancient humans is estimated to have been about 38 years, approximately half as long as modern humans and the average age of presentation of colorectal cancer is 50 years of age. How could this difference between the study groups have affected the data (if at all)?
Answer:
The short lifespan of ancient individuals should not affect our data, as our study focused on the germline variants, which by the nature doesn’t change during the entire life of the variant carriers, regardless of their age or cancer status.
Question:
The authors speculate (line 272) that pathogenic variations beneficial in promoting survival and reproduction. This statement should be justified as to how PVs can be beneficial.
Answer:
In the revision the following sentences have been included in Discussion:
Pathogenic variants are deleterious for better fitness. Therefore, evolution selection should suppress their presence in human population. However, multiple pathogenic variants were present at rather high prevalent level suggesting their potential benefit effects. While we don’t have enough evidence to support such possibility, we speculate that the pathogenic effects could be developmental stage-dependent that the pathogenic variants could be beneficial at the reproductive stage but deleterious at later reproduction stage. Alternately, the cancer caused by the pathogenic variants occurs mostly at later reproduction stage. Before reaching the stage, however, the pathogenic variants have already been transmitted to the next generation.
Round 2
Reviewer 2 Report
The Authors provided satisfying answers to the reviewer’s comments. A new fragment to discussion was added about limited number of ancient samples. It is obvious that the availability of sequenced human genomes provides easy to obtain material for analysis. The authors developed a methodological approach to the analysis evolutionary origin of germline pathogenic variations in different human genes. Subsequent works are created on the model of the first one and only the analyzed gene is changed.